# Potential impacts of climate change on geographical distribution of three primary vectors of African Trypanosomiasis in Tanzania's Maasai Steppe: *G. m. morsitans, G. pallidipes and G. swynnertoni*

**Happiness Jackson Nnko**[1]*, **Paul Simon Gwakisa**[2], **Anibariki Ngonyoka**[1], **Calvin Sindato**[3,4], **Anna Bond Estes**[5,6]

1 University of Dodoma, Dodoma, Tanzania, 2 Sokoine University of Agriculture, Morogoro, Tanzania, 3 National Institute for Medical Research, Tabora, Tanzania, 4 Southern African Centre for Infectious Disease Surveillance, Morogoro Tanzania, 5 Carleton College, Northfield, Minnesota, USA, 6 The Nelson Mandela African Institution of Science and Technology, Arusha, Tanzania

* happyjackson.nnko@gmail.com

**Data Availability Statement:** Due to a guideline that requires data to be shared upon specific

## Abstract

In the Maasai Steppe, public health and economy are threatened by African Trypanosomiasis, a debilitating and fatal disease to livestock (African Animal Trypanosomiasis -AAT) and humans (Human African Trypanosomiasis—HAT), if not treated. The tsetse fly is the primary vector for both HAT and AAT and climate is an important predictor of their occurrence and the parasites they carry. While understanding tsetse fly distribution is essential for informing vector and disease control strategies, existing distribution maps are old and were based on coarse spatial resolution data, consequently, inaccurately representing vector and disease dynamics necessary to design and implement fit-for-purpose mitigation strategies. Also, the assertion that climate change is altering tsetse fly distribution in Tanzania lacks empirical evidence. Despite tsetse flies posing public health risks and economic hardship, no study has modelled their distributions at a scale needed for local planning. This study used MaxEnt species distribution modelling (SDM) and ecological niche modeling tools to predict potential distribution of three tsetse fly species in Tanzania's Maasai Steppe from current climate information, and project their distributions to midcentury climatic conditions under representative concentration pathways (RCP) 4.5 scenarios. Current climate results predicted that *G. m. morsitans*, *G. pallidipes* and *G swynnertoni* cover 19,225 km², 7,113 km² and 32,335 km² and future prediction indicated that by the year 2050, the habitable area may decrease by up to 23.13%, 12.9% and 22.8% of current habitable area, respectively. This information can serve as a useful predictor of potential HAT and AAT hotspots and inform surveillance strategies. Distribution maps generated by this study can be useful in guiding tsetse fly control managers, and health, livestock and wildlife officers when setting surveys and surveillance programs. The maps can also inform protected area managers of potential encroachment into the protected areas (PAs) due to shrinkage of tsetse fly habitats outside PAs.

request, researchers who wish to access the data, can do so by making a request to the University of Dodoma; Directorate of Research, Publication and Consultancy at drpc@udom.ac.tz.

**Funding:** The author(s) received no specific funding for this work.

**Competing interests:** The authors have declared that no competing interests exist.

## Author summary

Spatial variation of African Trypanosomiasis burden depends on distribution of biotopes necessary for tsetse flies to thrive. Therefore, mapping the occurrence of the tsetse fly species is a useful predictor of African Trypanosomiasis transmission risk areas. Climate is a major determining factor for occurrence and survival of tsetse flies, the vector responsible for both HAT and AAT. Since resources for prevention and control of tsetse fly species and the disease they transmit are generally scarce in endemic settings, understanding potential impacts of climate change on tsetse fly species distribution in space and time is essential for informing coherent strategies for vector and disease control at a local scale.

## Introduction

Most climate change predictions show an upward trend in temperature for at least the next nine decades [1], but there is uncertainty with different climate models predicting different magnitudes of warming. On average, global temperature is expected to rise by 0.8–2.6˚C and by 1.5–3˚C in Africa by the year 2050 [2]. Such increases have potential to cause species habitat modification including range expansion or contraction in addition to altering their relationships with the bio-physical environment. The influence of climate change on species distribution is supported by evidence from fossil records [3] and observed trends from the twentieth to twenty-first centuries on species range shifts. For example, it is estimated that a change in 1˚C will lead to range shifts of 160km of the ecological zones on earth, implying that if the globe will warm by 3˚C by the year 2100, the flora and fauna of the North Pole will move approximately 480 km northward to remain within their thermal tolerances [4–5]. Some species of butterflies in Europe have been reported to shift further north as those zones become more habitable [6–8]. Predicted rise in temperature is also expected to transform dynamics of vector-borne diseases including African Trypanosomiasis, either by altering the vectors' and pathogens' geographical range, or their development and mortality rates [9–12].

Tsetse flies occur in Sub-Saharan Africa and their distribution is influenced by climate, vegetation and hosts. Climate, particularly temperature, is considered a major driver as it influences all others factors that determine tsetse occurrence. Trypanosomiasis remains a debilitating and fatal disease to livestock and humans, if left untreated. For instance, trypanosomiasis in livestock causes loss of over 4 billion USD due to 70% reduction of cattle density, 50% reduction in dairy and meat sales, 20% reduction in calving rates, and 20% increases in calf mortality in Sub-Saharan Africa [13]. In Tanzania, tsetse flies occur in over 65% of rangeland savannah ecosystems [14], exposing about 4 million people in rural communities to the risk of sleeping sickness and causing loss of approximately eight million USD annually due to nagana (AAT) induced low livestock productivity [15–17]. Since dynamics of African Trypanosomiasis is a function of tsetse fly competence, and the ecology and behavior of available hosts, spatial variation of disease burden depending on the distribution of biotopes necessary for tsetse flies to thrive is expected.

Trends in climate change and associated socioeconomic transformation is anticipated to continue altering tsetse fly habitats in Tanzanian rangelands. Nonetheless, empirical evidence to support the assertion about change in tsetse fly species distribution as a result of climate change is lacking in the country. Also, information that could aid tsetse control planning for future preparedness is rare to find in the country and absent at local scales. In the Maasai Steppe, for instance, knowledge on tsetse fly spatial variation is often based on old and coarse data and not publicly available.

Various scientific approaches have been used to understand the potential impacts of climate on spatial and temporal distribution of disease vectors. Some of the approaches include climate envelope models and correlations between climatic variables and vectors [18–21]. Climate envelopes are species distribution models that use climate data to define climate suitability for species to occur [22]. Specifically, these models rely on statistical correlations between species distributions points and their associated climate parameters to define a species' envelope of tolerance around existing ranges, thereby delineating a 'climate envelope' within which species thrive [19,22]. Compared to mechanistic models, climate envelope models do not incorporate data other than occurrence and environmental related data; so they do not predict fitness variation across climate gradients [23].

There has been research that studied risk of African Trypanosomiasis and tsetse fly burden in the Maasai Steppe [24–31]. However, none of these established potential impacts of climate change on distribution of tsetse. To fill this gap, a general question on what is the potential impact of climate change on the distribution of common *Glossina* species found in the study area was investigated. This study adopted the general definition of climate envelopes in which models were built using climate variables to define areas that have suitable climate for tsetse flies and model their distribution based on current climate under which they have been observed. Prediction for future distribution was carried out to understand how African Trypanosomiasis transmission hotspots might change under future climate scenarios. This information may help stakeholders to allocate scarce resources in preventing African Trypanosomiasis by implementing more targeted interventions. This study also may form a basis for a large national and regional scale prediction of future African Trypanosomiasis transmission hotspots.

## Methodology

### Study area

This study was carried out in the Tanzanian Maasai Steppe, located between 1.5 to 5° South latitude and 35 to 37° East Longitude (Fig 1). It covers an area of more than 60,000 km$^2$ with a population of over 600,000 people, mainly practising pastoralism and to a lesser extent, agro-pastoralism and tourism [32]. The region is semi-arid and a human-wildlife-livestock system, receiving up to 500 mm of rainfall per annum. Rainfall patterns dictate movement of pastoralists and their herds and wildlife in search for water and pastures. These movements increase the likelihood of disease transmission between domestic animals, people and wildlife [3].

### Data collection

**Species occurrence and background data.** This study targeted three *Glossina* species *G. m. morsitans*, *G. pallidipes* and *G. swynnertoni* commonly found in the Maasai Steppe [27–29]. Abundance data were collected through entomological field surveys carried out once in the dry season, November 2015 and once in the wet season, May 2016. A total of 99 baited epsilon traps [33] were placed in Simanjiro and Monduli districts. Traps were deployed in stratified random subsampling of the major vegetation types [34] at a distance of at least 200m apart [16,33]. At each trapping site, numbers of tsetse flies caught and geographical coordinates were recorded using hand-held Global Positioning System (GPS). The collected abundance data were converted to presence data for each of the GPS locations, yielding a total of 32, 59 and 29 unique occurrence points for *G. m.morsitans*, *G. pallidipes* and *G. swynnertoni*, respectively, after eliminating duplicate records resulting from multiple entries for a particular season. Duplicate records were removed using ecological niche modelling tools (ENMTools) software version 1.4.3 [35]. The occurrence data were used with climate predictor variables as

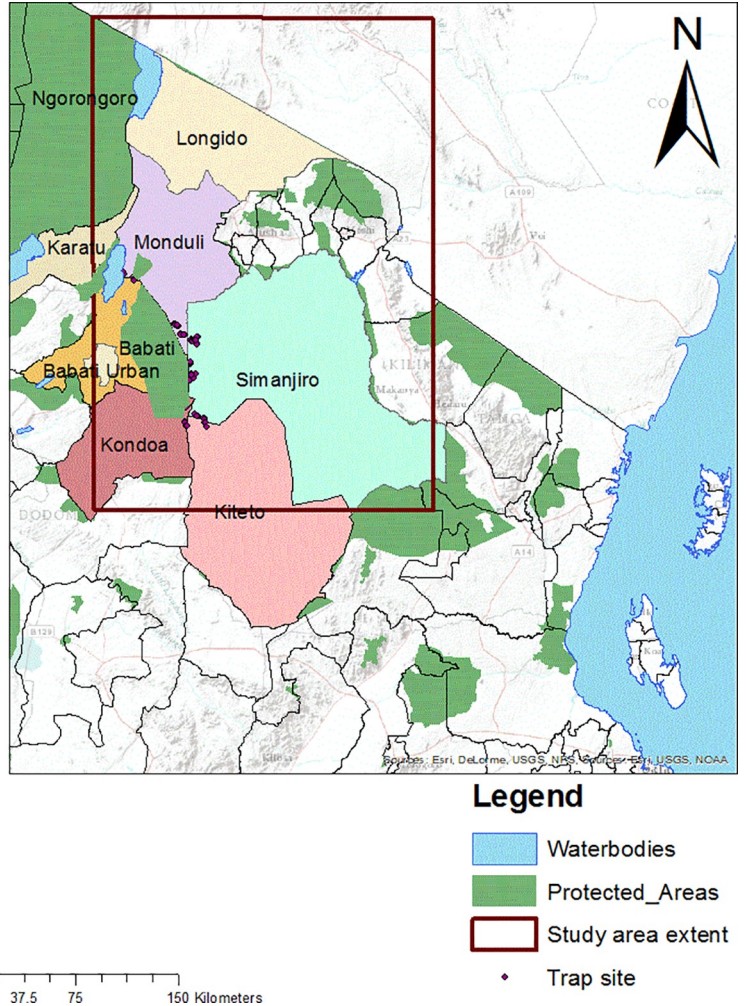

**Fig 1. An extract of a map of Tanzania showing the study districts (Kiteto, Longido, Monduli and Simanjiro) forming the Maasai Steppe.**

input in MaxEnt (v 3.3.3k) [36], to create climate envelope models for the three species. MaxEnt is a species distribution model developed to work with presence-only data, and has been widely used in modelling the probability of occurrence of species across space and time in areas that have not been sampled [36–38]. Since dispersal of tsetse flies is dependent on availability of suitable hosts, and the study area is home to numerous hosts (wildlife and livestock), the study assumed that all districts of the Maasai Steppe had potential for attracting tsetse flies. For this reason, background data were sampled from the whole study area [38–40].

**Climate layers.** Predictive models for tsetse fly species distribution were made using the occurrence data and current climate variables (Table 1). The initial candidate layers considered in the model were elevation, precipitation of the wettest month (April), mean maximum temperature of the warmest month (February), mean maximum temperature of the driest month (September) and mean minimum temperature of the coldest month (July). Both maximum and minimum temperature affect tsetse fly activity patterns and play an important role in determining the development of tsetse flies [41–42] and differentiation and proliferation of trypanosomes and thus trypanosomes prevalence [27,43]. On the other hand, precipitation affects tsetse fly development indirectly by maintaining vegetation "*tsetse fly habitats*" [44].

**Table 1. Candidate covariates tested used in initial model runs, and the bolded ones used in the best-performing MaxEnt models.**

| Variable | Type | Units | Resolution | source |
|---|---|---|---|---|
| **Precipitation of the wettest month (April)** | **Continuous** | **ml** | **833.33m** | **http://www.worldclim.org** |
| **Mean maximum temperature of the warmest month (April)** | **Continuous** | **⁰C*10** | **833.33m** | **http://www.worldclim.org** |
| **Mean minimum temperature of the coolest month (July)** | **Continuous** | **⁰C*10** | **833.33m** | **http://www.worldclim.org** |
| **Altitude/elevation** | **Continuous** | **msl** | **833.33m** | **http://www.worldclim.org** |
| Mean maximum temperature of the driest month (September) | Continuous | ⁰C*10 | 833.33m | http://www.worldclim.org |

Apart from maintaining vegetation and humidity for tsetse fly to thrive, precipitation also affects tsetse fly species indirectly by causing local flooding which may drown pupae that are buried in loose soil [45], and so it was included in predictor variables. In addition, a number of reports have indicated fluctuation of tsetse fly abundance during rainy season [25,27,46,47]. Elevation, which is a proxy for temperature, was also used as a predictor variable in order to gain insight regarding the potential altitude limit for tsetse fly species to thrive. Although land cover/use and density of animals also influence tsetse fly distribution in space and time [16], this information was not included in the study due to inconsistency of available data. Models created using current climate variables were mapped on to future climate layers to understand how changing climate might influence tsetse distribution and thereby African Trypanosomiasis transmission risk. For the future climate projection scenario (year 2050), this study used 833.33m resolution Coupled Model Inter-comparison Project (CMIP5) global circulation model (GCM).

Of the many possible GCMs to use, CMIP5 was chosen because the CMIP5 models are relatively more advanced (fine-tuned) and they use RCP scenarios compared to previous GCMs that were released in or before 2010. In particular, the climate system model from Beijing Climate Center (BCC-CSM1-1) was used and the RCP 4.5 was selected for this study. The BCC-CSM1 was chosen for this analysis because it is among the models that have been suggested to capture the key processes relevant to our study area [48]. Although there is uncertainty associated with any future climate scenario, these data provide reasonable predictions that can be useful for planning.

**Modelling procedures.** In order to minimize the use of correlated variables that may mask the contribution of individual variables and cause difficulties in results interpretation [37,49], pairwise collinearity tests of predictor variables was performed using ENMTool 1.4.3 [35–36]. Temperature variables and altitude were highly correlated but mean minimum and maximum temperature of coldest and warmest month respectively were retained because of their high biological relevance to tsetse fly species [46]. Altitude was also included in the model to gain insights regarding the elevation limits of tsetse fly species distribution. Mean maximum temperature of the driest month was omitted from analysis because of the relatively lower knowledge of effects of dryness on tsetse fly survival and development except when it is accompanied by other parameters such as temperature and precipitation which were already candidate variables.

MaxeEnt was used to model the probability of species occurrence based on unique occurrence points [36–38]. A sample bias file was excluded from the model with the assumption that tsetse flies are likely to be present in a large part of the study area due to wide distribution of hosts [17,50]. Because there were more than 15 occurrence points, MaxeEnt was run using linear, quadratic and hinge features [51]. The model was set to run with 500 iterations and 10 replicates with default parameters for regularization and the jackknife estimates (measure of variable influence).

**Model assessment.** Four variables were included in MaxEnt along with the occurrence data. An initial SDM was run in MaxEnt (one run; raw output setting) to acquire lambda

values used in ENMTools v.1.4. 3 [35] to calculate Akaike's Information Criterion (AICc; AIC) and Bayesian Information Criterion (BIC) [52] for a model fit with four, three and two variables, respectively (Table 2). This method selects the most parsimonious model. The model that was most parsimonious in this study (lowest AIC, AICc, BIC and high area under the receiver operating curve (AUC) value) had all four variables. The best model for each species was validated using 10-fold cross-validation, with the averages of 10 model runs representing the final output. Model performance as well as the contribution of predictor variables were assessed by using AUC, and variable importance was assessed using the relative gain contribution of each variable and jackknife tests compared using AUC, test gain and regularized training gain. Marginal and single variable response curves were used to depict the relationship between tsetse fly species and predictor variables. Final outputs included predictive maps of the probability of tsetse fly species presence based on climate suitability. The probability scores (numeric values between 0 and1) were displayed in ArcGIS 10.5 to show the current and future habitat suitability for each of the three tsetse fly species.

## Results

### Model selection

The distribution models for each tsetse fly species performed better than base/random (AUC>0.5) (Table 2). The model that included all four predictor variables had the best fit (Table 2). The results presented in all subsequent sections are based on this model.

### Variable contribution and climate suitability map for *G. m. morsitans*

Altitude accounted for more than one third (35.1%) of the variation in the climate suitability model for *G. m. morsitans* occurrence, followed by precipitation of the wettest month (32.1%), maximum temperature of the warmest month (22.3%), and minimum temperature of the coldest month (10.6%). Based on the 10 percentile training presence logistic threshold (10% minimum threshold), the model showed that currently 32% (19,225 km$^2$) of the entire Maasai Steppe ($\approx$ 60,000 km$^2$) has suitable climate for *G. m. morsitans* (Fig 2), but this would shrink to 7.4% (4,447.34 km$^2$) by 2050' (Fig 3).

**Table 2. Model performance based on AUC, AIC, AICc and BIC values for tsetse fly species occurrence and different combinations of the environmental variables.**

| Species | Model assessment | Tmax of warmest month<br>Tmin of coldest month | Precipitation of the wettest month<br>Tmax of warmest month<br>Tmin of coldest month | Altitude<br>Precipitation of the wettest month<br>Tmax of warmest month<br>Tmin of coldest month |
|---|---|---|---|---|
| *G.m.morsitans* | AUC | 0.850 | 0.902 | 0.938 |
| | AIC | 702.78 | 667.27 | 625.79 |
| | AICc | 709.04 | 680.47 | 642.21 |
| | BIC | 714.51 | 683.39 | 643.38 |
| *G. pallidipes* | AUC | 0.818 | 0.919 | 0.959 |
| | AIC | 1302.59 | 1198.26 | 1108.75 |
| | AICc | 1304.79 | 1202.85 | 1115.54 |
| | BIC | 1317.13 | 1219.04 | 1133.68 |
| *G. swynnertoni* | AUC | 0.840 | 0.854 | 0.899 |
| | AIC | 624.99 | 614.66 | 576.83 |
| | AICc | 630.32 | 626.88 | 601.09 |
| | BIC | 634.56 | 628.33 | 594.60 |

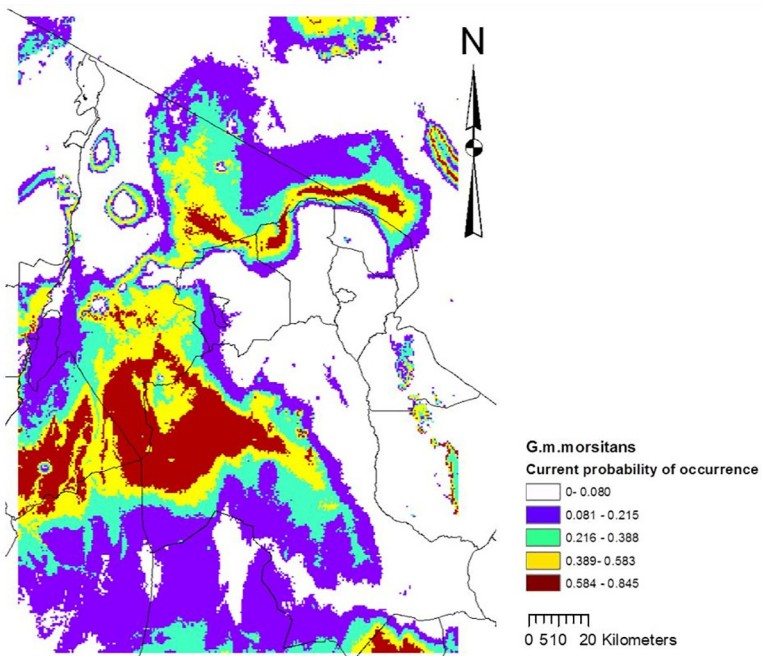

**Fig 2. Current climate suitability map for the best performing model with the *G.m. morsitans* occurrence data, and all 4 environmental variables: elevation, precipitation of the wettest month (April), mean maximum temperature of the warmest month (February), and mean minimum temperature of the coldest month (July).**

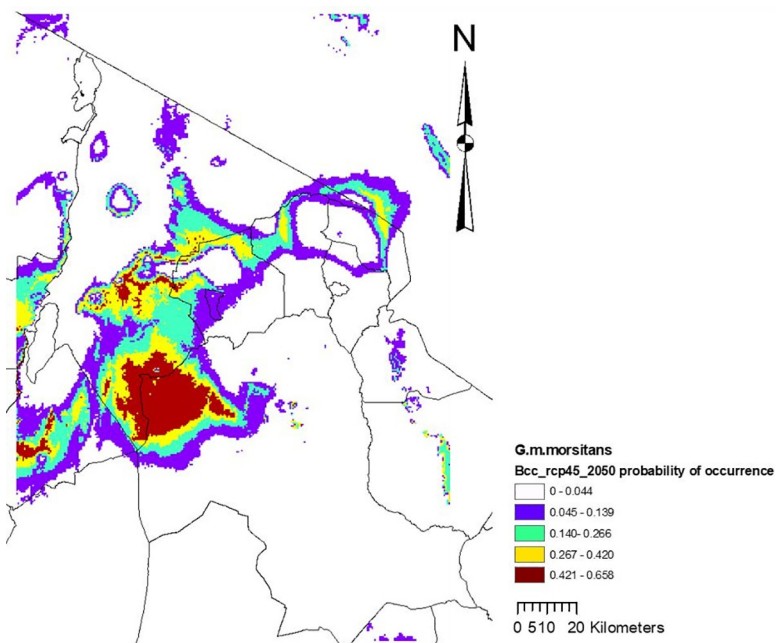

**Fig 3. Midcentury (2050) climate suitability map for the best performing model with the *G.m. morsitans* occurrence data, and all 4 environmental variables: elevation, precipitation of the wettest month (April), mean maximum temperature of the warmest month (February), and mean minimum temperature of the coldest month (July).** In these figure we see that the probability of occurrence decreases with time (comparing current and midcentury) from the maximum values of 0.845 to 0.658, with contracted habitat.

Variable response curves indicated that the probability of occurrence of *G. m. morsitans* drops off dramatically above 1000m of altitude when all variables are included in the model, but a very peaked response to altitude ≈1200msl (*S1A Fig*) and almost no probability of occurrence above 2500msl when that is the only variable considered (*S2A Fig*). Marginal and single variable response curves were however similar for precipitation of the wettest month, showing a preference (probability of presence ≥ 0.6) for precipitation between 140-230mm per month and almost no chance of occurrence below 100mm/month or above 350mm/month (*S1B* and *S2B* Figs). The probability of occurrence of *G. m. morsitans* drops off dramatically above 28˚C maximum temperature when all variables are included in the model (*S1C Fig*), but a peaked response to maximum temperature of ≈ 28˚C for the mean maximum temperature of the warmest month, with minimal chances of occurrence below ≈ 15˚C or above ≈ 32˚C maximum temperature values when used as the only variable in the model (*S2C* Fig).

The probability of occurrence of *G. m. morsitans* drops off dramatically above 14˚C minimum temperature when all variables are included in the model, reaching the peak response at a minimum temperature of ≈ 13˚C for the mean minimum temperature of the coldest month (*S1D Fig*), with rare chances of occurrence below 0˚C or above 16˚C minimum temperature when used as the only variable (*S2D Fig*). Altitude accounted for over one third of variation in the climate suitability model for *G. m. morsitans* occurrence but maximum temperature of the warmest month provided the best fit to the training model when used in isolation, while precipitation of the wettest month appeared to have the most information that is not captured by other variables and thus decreases the gain the most when omitted (S3 Fig).

## Variable contribution and climate suitability map for *G. pallidipes*

Precipitation of the wettest month accounted for almost two-thirds (60.4%) of the variation in habitat suitability, followed by altitude (23.0%) and maximum temperature of the warmest month (16.6%). Based on the 10 percentile training presence logistic threshold, the model showed that current suitable habitat for *G. pallidipes* covers 11% (7113 km$^2$) of the Maasai Steppe (Fig 4) and by 2050, the model indicated only 918 km$^2$ with suitable habitat for this species (Fig 5).

Variable response curves indicated that the probability of occurrence of *G. pallidipes* drops off dramatically above 1,000m of altitude when all variables are included in the model, reaching its peak response at altitude ≈1,200msl (*S4A Fig*) and almost no probability of occurrence above 3,000msl when that is the only variable considered in the model (*S5A Fig*). Marginal and single variable response curves were similar for precipitation of the wettest month, showing a preference (probability of presence ≥ 0.6) for precipitation between 140-180mm per month (*S4B Fig*), and almost no chance of occurrence below 120mm/month or above 330mm/month *S5B Fig*). The probability of occurrence of *G. pallidipes* drops off dramatically above 28˚C maximum temperature when all variables are included in the model (*S4C Fig*), and a peak response was observed at a maximum temperature of ≈ 28˚C for the mean maximum temperature of the warmest month, and almost no chance of occurrence below 10˚C or above 34˚C maximum temperature when used as the only variable (*S5C Fig*). The probability of occurrence of *G. pallidipes* drops off dramatically above 10˚C minimum temperature when all variables are included in the model (*S4D Fig*), but a very peaked response to minimum temperature of ≈ 13˚C for the mean minimum temperature of the coldest month and almost no chance of occurrence below -5˚C or above 17˚C minimum temperature when used as the only variable (*S5D Fig*).

Precipitation of the wettest month provided the best fit to the training data when used in isolation and best predicted the distribution of the *G. pallidipes*. This variable also appears to have the most information that is not present in the other variables, as it decreases the gain the most when it is omitted (S6 Fig).

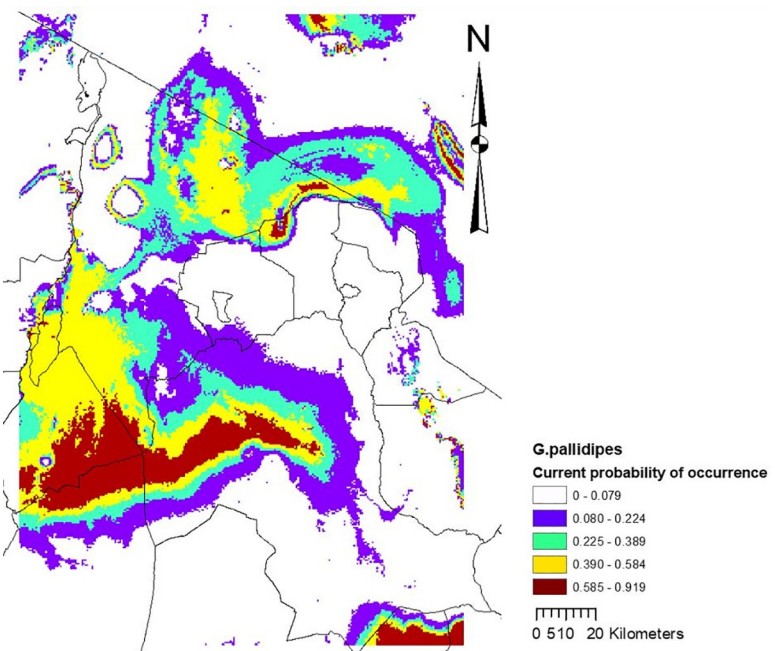

**Fig 4. Current climate suitability map for the best performing model with the *G. pallidipes* occurrence data, including all 4 variables.**

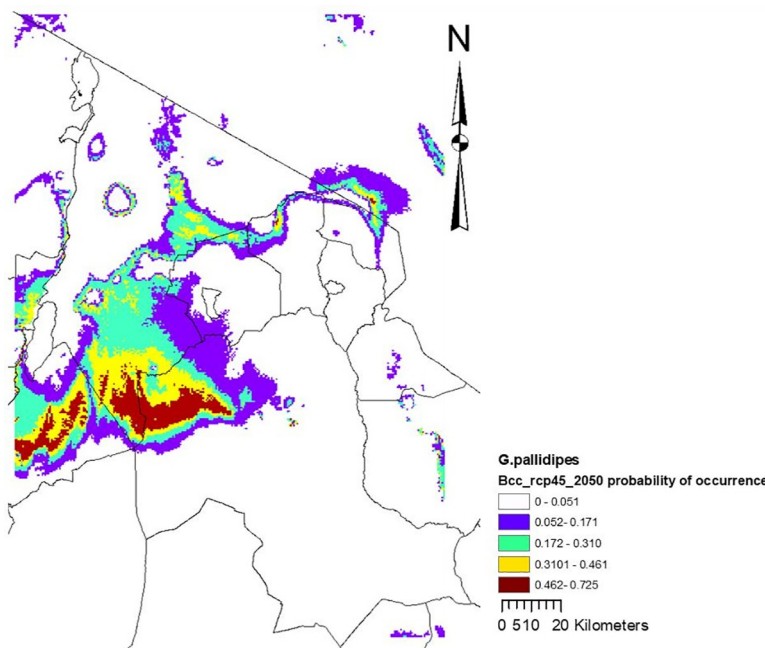

**Fig 5. Midcentury (2050) climate suitability map for the best performing model with the *G. pallidipes* occurrence data, including all 4 variables.** In these maps we see that the probability of occurrence decreases with time (comparing current and midcentury) from the maximum values of 0.919 to 0.725, with shrunk habitat.

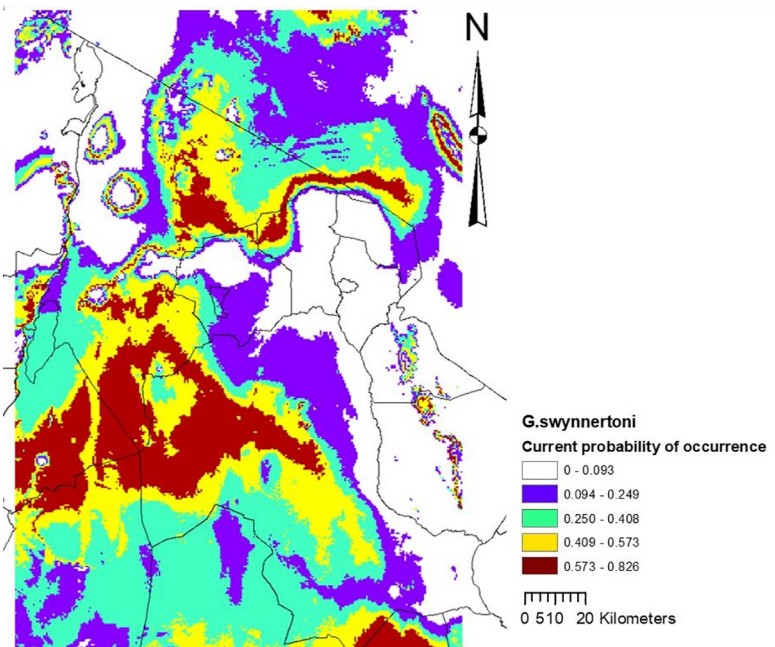

**Fig 6. Current climate suitability map for *G. swynnertoni*, for the model including all four predictor variables.**

## Variable contribution and climate suitability map for *G. swynnertoni*

Altitude contributed almost half (47.5%) of the variation in climate suitability for *G. swynnertoni* occurrence, followed by precipitation of the wettest month (27.4%), minimum temperature of the coldest month (22%), and maximum temperature of the warmest month (3.1%). Based on the 10 percentile training presence logistic threshold, it was revealed that current suitable climate for *G. swynnertoni* covers 32,335 km$^2$ (Fig 6), but is predicted to shrink to 7,374 km$^2$ by the year 2,050 (Fig 7).

Variable response curves indicated that the probability of occurrence of *G. swynnertoni* drops off dramatically above 1000m of altitude when all variables are included in the model (S7A Fig) but a very peaked response to altitude ≈1300msl and almost no probability of occurrence above 2500msl when that is the only variable considered (S8A Fig). Variable response curves indicated that the probability of occurrence of *G. swynnertoni* drops off dramatically above 140mm of rainfall when all variables are included in the model (S7B Fig), but a very peaked response to precipitation ≈160mm for the precipitation of the wettest month and almost no probability of occurrence above 400mm/month or below 90mm/month when that is the only variable considered (S8B Fig). The probability of occurrence of *G. swynnertoni* drops off dramatically above 28˚C maximum temperature when all variables are included in the model (S7C Fig), but a peaked response to maximum temperature of ≈ 28˚C for the mean maximum temperature of the warmest month, and almost no chance of occurrence below 10˚C or above 34˚C maximum temperature when used as the only variable (S8C Fig).

The probability of occurrence of *G. swynnertoni* drops off dramatically above 14˚C minimum temperature when all variables are included in the model (S7D Fig). The peak probability was observed at a minimum temperature of ≈ 14˚C for the mean minimum temperature of the coldest month with reduced chances of occurrence below 0˚C or above 16˚C minimum temperature when used as the only variable (S8D Fig).

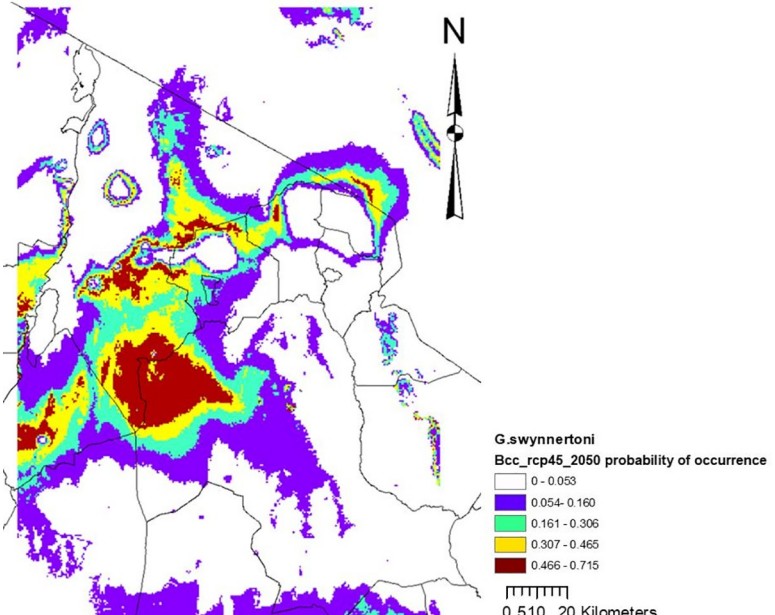

**Fig 7. Midcentury (2050) climate suitability map for *G. swynnertoni*, for the model including all four predictor variables.** Similarly, these maps indicate that the probability of occurrence decreases with time (comparing current and midcentury) from the maximum values of 0.826 to 0.715, with narrowing habitat.

The best fit to the *G. swynnertoni* training data was provided by altitude when used by itself. Altitude indicated the best fit to the test data and best predicted the distribution of the *G. swynnertoni* test data. Also, omission of this variable decreased the gain the most, meaning altitude had most information that is not present in other variables (S9 Fig).

## Discussion

Tsetse fly occurrence poses public health challenges and exacerbates economic hardships due to the investment needed to control tsetse flies and treat the diseases they transmit. Since climate is the dominant factor that determines tsetse fly occurrence, and the resources for controlling tsetse and trypanosomiasis are scarce, understanding how the changes in climate at the local scale affects the spatial and temporal distribution of tsetse fly species is critical in identifying the most likely vulnerable places, and better targeting limited resources. The SDM used in this study provides useful information for public health, livestock development stakeholders and wildlife managers to plan for future potential climates effects across space and time.

This study used MaxEnt species distribution modelling to understand the influence of altitude and climate variables on tsetse fly species occurrence, and make predictions about future distribution based on predictive climate models. The models yielded current and future potential climate distribution maps for *G. m. morsitans*, *G. pallidipes* and *G. swynnertoni*, and predicted an overall reduction in the area of the Maasai Steppe that will have suitable climate for the three *Glossina* species. Prediction also indicated the probability of these three tsetse fly species to inhabit relatively higher latitude by mid-century. Compared to current conditions, in the year 2050, area with suitable climate will decline to 23.13%, 12.9% and 22.8% of current suitable area for *G. m. morsitans*, *G. pallidipes* and *G. swynnertoni*, respectively. The reason for this could be explained by the temperature response curves, which indicated 34°C mean maximum temperature of the warmest month and 17°C mean minimum temperature of the coldest

month to be maximum upper and lower temperature thresholds for these three species. The range reduction across the Maasai Steppe can be attributed to future climates exceeding these thresholds whereby, by mid of the century, maximum temperature is expected to have risen by 1.7˚C in the Maasai Steppe [48]. The temperature thresholds that limit tsetse fly distribution and abundance have also been shown in other studies from the Maasai Steppe, based on intensive longitudinal sampling over smaller geographic areas [27]. These observations complement the suggestion that climate change in some parts of East Africa would result in overall reduction of habitat suitability range for tsetse flies, but also a spread out of suitable range particularly in high-altitude areas that currently are less suitable for the species due to low temperatures [18]. Hulme also predicted a contraction of *G. m.morsitans* geographic range owing to climate change expected to affect the SADC region [53]. Influence of climate on the distribution of *Glossina* species has been explained in the previous studies [41,42,54–57] and *G. m.morsitans*, *G. pallidipes* and *G. swynnertoni* are among the groups of tsetse flies whose relative abundance tends to decrease with high temperature. Our model forecasts suitable area for all three species that will shrink in the Maasai Steppe by 2050 under RCP 4.5, suggesting populations of these species may crash or may adapt to increasing maximum temperatures by moving upward in elevation. In fact, the models predicted a suitable altitude for *G. m. morsitans*, *G. pallidipes* and *G. swynnertoni* from around 1,000msl currently observed, to around 2,500m, 3,000m and 2,500m elevation, respectively, indicating these species may become problematic in high altitude ecosystems of the study area, if other ecological requirements for these species will be met in those habitats.

The importance of the four variables that were selected through our parsimony analysis to the ecology of the three *Glossina* species indicates the importance of careful scrutiny of available environmental data for a study site of interest. Although there was variation in variable contribution to specific species models, mean maximum temperature of the warmest month and mean minimum temperature for the coldest month indicated similar response curves. Specifically, mean maximum temperature of the warmest month, and mean minimum temperature of the coldest month have relevant ecological importance to the distribution of tsetse fly species. For example, the logistic probability response curves indicated higher maximum temperature of the warmest month and higher minimum temperature of the coldest month decreases likelihood of all three *Glossina* species occurrence, likely because both low and high temperatures affect development of all three tsetse species at various life stages [41]. Effects of hotter and colder environments on various developmental stages of tsetse fly species has also been reported [58–59].

Logistic probability response curves indicated that higher precipitation during the wettest month decreases the likelihood of occurrence of the three *Glossina* species considered in this study. Generally, no record is known on the direct effect of rainfall on tsetse flies, but it is thought that high rainfall may cause local flooding which may wash out pupae that are buried in loose soil, leading to tsetse fly depopulation and thus low probability of occurrence. Although responses to this variable indicated similar trends in all three species, the importance of the variable in models for the different species varied dramatically. For example, precipitation of wettest month contributed 60.4% of the relative gain to the *G. pallidipes* model and provided the best fit to the model, indicating that the species can respond differently to the climate variables. In particular, precipitation in the wettest month may be more important to the distribution of *G. pallidipes* owing to the species' ecology. *G. pallidipes* is strongly associated with wetter habitats, and so relatively hydrophilic, unlike *G. m. morsitans* and *G. swynnertoni*.

In all three tsetse fly species models, altitude had a relatively high contribution to the model gain, but did not necessarily provide the best fit to the training model. For example, altitude

contributed 35.1% of relative gain to the *G. m. morsitans* model and 23% for *G. pallidipes* respectively. However, the best fit to the training models for these two species were provided by mean maximum temperature of the warmest month and precipitation of the wettest month. This may be because temperature and rainfall have more biological relevance to tsetse flies compared to altitude. Although altitude indicated high contribution (47.5%) to the *G. swynnertoni* model and also had the best fit, it should however be noted that all occurrence points were obtained at relatively lower altitudes and this might have influenced the results. Nevertheless, all *Glossina* species responded similarly to altitude, with response curves for all species indicating low preference for higher altitude. This is because higher altitudes are characterized by lower temperature that affects tsetse fly development [42]. Given that altitude and temperature were highly correlated, it was initially considered that by including altitude in the model, it could have masked the contribution of variables with greater biological relevance [37]. However, because relationships between tsetse flies and temperature are well-established [41,42,60,61], altitude was included in the models in order to gain insight into how tsetse fly species are likely to expand their range to higher elevations under future increases in temperature.

Extrapolated over larger areas, our findings could indicate either increases or decreases in suitable tsetse range. Likewise, predictions of climate impacts of tsetse distribution in Africa do not all agree. Some studies have suggested that climate change in some parts of East Africa would result in a spreading out of suitable range for tsetse flies particularly in high-altitude areas that currently exclude the species due to low temperatures, but also there is a chance of range contraction of tsetse flies in some locations [18]. Other reports have suggested a decline in the distributional range of tsetse fly species owing to climate change. Furthermore, it should be noted that climate variables are not the sole predictors of future tsetse distribution. Other factors such as host availability and suitable vegetation will also influence where tsetse are found, but are more difficult to model into the future. Distribution maps based on relationships with climate variables can therefore be considered to be maximum potential distributions.

Although the findings of this study are based on only a single GCM model, BCC-CSM1-1 from CMIP5, it is considered to have better predictive capacity because it uses RCP and is at a relatively finer resolution of about 1km. The fact that these findings agree with previous findings reported by Hulme [53] and Rodgers and Randolph [54] that used relatively older GCM versions, increases the confidence that climate is more likely to push distribution of tsetse flies into new areas, while removing it from others. For this reason, maps produced by this study can improve the efficiency and lower the cost of future surveillance. Also, the methods employed by this study can be adopted to generate high resolution species distribution maps under current and future climate scenarios for larger areas and for other vectors that pose threats to both public health and economic development. Tsetse fly control managers can incorporate the maps created from these models into integrated pest management regimes, and further tailor them based on what is already known about the Maasai Steppe. Finally, maps such as these may be displayed to the public to increase awareness of climate change implications in the Maasai Steppe and other areas that are tsetse infested. These maps can as well inform protected areas managers of the likely encroachment due to shrinkage of tsetse fly habitats even in protected areas.

Limitations of this study include the fact that the study approach was climate envelope models which does not predict the expected ability/fitness of tsetse flies to adapt to the climate change. Inclusion of other ecological requirement variables would improve the prediction of general habitat suitability other than only climate suitability.

## Supporting information

**S1 Fig. Marginal response curves for the best performing model with *G.m.morsitans* occurrence data.** Temperature is reported in 0C * 10.
(TIF)

**S2 Fig. Single variable response curves for the best performing model with *G.m.morsitans* occurrence data.** Temperature is reported in 0C * 10.
(TIF)

**S3 Fig. Jackknife estimates of variable importance for the best-performing model for *G.m. morsitans*.** Variable performance is assessed via the variables' impact to training and test gain (top and middle) and AUC (bottom).
(TIF)

**S4 Fig. Marginal response curves for the best performing model with *G.pallidipes* occurrence data.** Temperature is reported in $^0$C * 10.
(TIF)

**S5 Fig. Single variable response curves for the best performing model with *G.pallidipes* occurrence data.** Temperature is reported in $^0$C * 10.
(TIF)

**S6 Fig. Jackknife estimates of variable importance for the best-performing model for *G. pallidipes*.** Variable performance is assessed with training and test gain (top and middle) and AUC (bottom).
(TIF)

**S7 Fig. Marginal response curves for the best performing model with G.*swynnertoni* occurrence data.** Temperature is reported in $^0$C * 10.
(TIF)

**S8 Fig. Single variable response curves for the best performing model with G.*swynnertoni* occurrence data.** Temperature is reported in $^0$C * 10.
(TIF)

**S9 Fig. Jackknife estimates of variable importance for the best-performing model for *G. swynnertoni*.** Variable performance is assessed with training and test gain (top and middle) and AUC (bottom)
(TIF)

## Author Contributions

**Conceptualization:** Happiness Jackson Nnko, Paul Simon Gwakisa, Calvin Sindato, Anna Bond Estes.

**Data curation:** Happiness Jackson Nnko, Anibariki Ngonyoka.

**Formal analysis:** Happiness Jackson Nnko.

**Methodology:** Happiness Jackson Nnko, Anibariki Ngonyoka, Calvin Sindato, Anna Bond Estes.

**Supervision:** Paul Simon Gwakisa, Anna Bond Estes.

**Writing – original draft:** Happiness Jackson Nnko.

**Writing – review & editing:** Happiness Jackson Nnko, Anna Bond Estes.

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
