## [Decision Letter · Decision Letter 0]

30 Jul 2020

Dear Dr Nnko,

Thank you very much for submitting your manuscript "Potential Impacts of Climate Change on Geographical Distribution of Three Primary Vectors of African Trypanosomiasis in Tanzania Maasai Steppe; G. m. morsitans, G. pallidipes and G. swynnertoni" for consideration at PLOS Neglected Tropical Diseases. As with all papers reviewed by the journal, your manuscript was reviewed by members of the editorial board and by several independent reviewers. In light of the reviews (below this email), we would like to invite the resubmission of a significantly-revised version that takes into account the reviewers' comments. 

We cannot make any decision about publication until we have seen the revised manuscript and your response to the reviewers' comments. Your revised manuscript is also likely to be sent to reviewers for further evaluation.

Sincerely,

Alvaro Acosta-Serrano

Deputy Editor

Alvaro Acosta-Serrano

Deputy Editor

Reviewer's Responses to Questions

**Key Review Criteria Required for Acceptance?**

**Methods**

-Are the objectives of the study clearly articulated with a clear testable hypothesis stated?

-Is the study design appropriate to address the stated objectives?

-Is the population clearly described and appropriate for the hypothesis being tested?

-Is the sample size sufficient to ensure adequate power to address the hypothesis being tested?

-Were correct statistical analysis used to support conclusions?

-Are there concerns about ethical or regulatory requirements being met?

Reviewer #1: (No Response)

Reviewer #2: The objective of the study is clearly articulated with a clear testable hypothesis stated and the study design is appropriate, population described. The statistical analysis supports the conclusion, but use of a single wet season and a single dry season may not be enough to make proper conclusions for such a study. One season may not give representative data. It would be good if they could consider data for atleast 3 wet seasons and 3 dry seasons.

**Results**

-Does the analysis presented match the analysis plan?

-Are the results clearly and completely presented?

-Are the figures (Tables, Images) of sufficient quality for clarity?

Reviewer #1: (No Response)

Reviewer #2: Yes

**Conclusions**

-Are the conclusions supported by the data presented?

-Are the limitations of analysis clearly described?

-Do the authors discuss how these data can be helpful to advance our understanding of the topic under study?

-Is public health relevance addressed?

Reviewer #1: (No Response)

Reviewer #2: Yes

**Editorial and Data Presentation Modifications?**

Reviewer #1: (No Response)

Reviewer #2: I have suggested published articles of previous work in this area for the authors to read before making certain statements, details are in the manuscript (attached). These are minor revisions in the abstract and methods.

**Summary and General Comments**

Reviewer #1: The study outlined in the manuscript deployed MaxEnt species distribution modelling (SDM) and ecological niche modelling tools to depict likely distribution patterns of three G. m. morsitans, G. pallidipes and G. swynnertoni tsetse fly species in Tanzania Maasai Steppe based on the current climate profile and predicted change by mid-century under Representative Concentration Pathway (RCP) 4.5 scenario. The results predict that habitable areas may decrease by up to 23.13%, 12.9% and 22.8% for the three species, respectively, by 2050.

The study appears to have been well conducted. However, it requires significant editorial correction or improvement, and specially the Discussion part, which appears to be too wordy and thematically repetitive.

Some corrections and changes are indicated below.

Line 66: ‘dairy’, not ‘diary’.

Line 107: ‘practising’ instead of ‘practicing’?

Lines 127-30, 2nd part of the sentence: change the second part of the sentence from ‘including to predict the probability of occurrence of species across space and time..’.

Line 140: change ‘plays’ to ‘play’.

Line 141: ‘…..since blood meals are the only tsetse fly nutrition…….’.

Line 209-213: ‘…… the model showed that currently 32% (19,225 km2) of the entire Maasai Steppe (≈ 60,000 km2) has suitable climate for G. m. morsitans, but this would shrink to 7.4% (4,447.34 km2) by 2050’.

Line 258-262: The three sentences appear to repeat more or less the same message……it can be condensed into one or two sentences.

Reviewer #2: The work is fit for publication if they read the articles i have suggested and make the necessary adjustments.

PLOS authors have the option to publish the peer review history of their article (what does this mean?). If published, this will include your full peer review and any attached files.

Reviewer #1: No

Reviewer #2: No
---

## [Decision Letter · Decision Letter 1]

16 Dec 2020

Dear Dr Nnko,

We are pleased to inform you that your manuscript 'Potential Impacts of Climate Change on Geographical Distribution of Three Primary Vectors of African Trypanosomiasis in Tanzania Maasai Steppe; G. m. morsitans, G. pallidipes and G. swynnertoni' has been provisionally accepted for publication in PLOS Neglected Tropical Diseases.

Best regards,

Paul O. Mireji, PhD

Associate Editor

Alvaro Acosta-Serrano

Deputy Editor

Reviewer's Responses to Questions

**Key Review Criteria Required for Acceptance?**

**Methods**

-Are the objectives of the study clearly articulated with a clear testable hypothesis stated?

-Is the study design appropriate to address the stated objectives?

-Is the population clearly described and appropriate for the hypothesis being tested?

-Is the sample size sufficient to ensure adequate power to address the hypothesis being tested?

-Were correct statistical analysis used to support conclusions?

-Are there concerns about ethical or regulatory requirements being met?

Reviewer #1: Yes, the objectives of the study are clear. The study design based on Beijing Climate Center (BCC-CSM1-1) and and and the RCP 4.5 stabilization scenario are appropriate. Given it focus on climate based ecological niche modeling, there are no concerns about ethical or regulatory requirements.

Reviewer #3: The authors have clearly set out the study and provided an informative way of predicting what may happen to the three species of tsetse flies when the natural habitat is influenced by habitat. The design looks at a point in time and uses the point in time to estimate what the situation may be like in another point in time in the future based on a few selected factors. The models used present the possibility for estimating the distribution of the three species. however the weak point is that the study was based on 2 seasons and the prediction is based on only one year of study. The sample size may be based on a period when the fly behaviour was modified and may not have been a true representation of the conditions. I however feel the paper's intention was to demonstrate the feasibility of use of the methodology and more accurate estimates may be calculated based on a larger study period. I accept the paper as a demonstration of the technology.

**Results**

-Does the analysis presented match the analysis plan?

-Are the results clearly and completely presented?

-Are the figures (Tables, Images) of sufficient quality for clarity?

Reviewer #1: The results presented match the modeling protocol used.

Reviewer #3: Despite the short study period the results were analysed in an exciting manner and are clearly presented with appropriate figures for emphasis. Using the data available the authors have made a good job of the analysis. The figures look clear but if there is a way of making them clearer that would benefit the reader more

**Conclusions**

-Are the conclusions supported by the data presented?

-Are the limitations of analysis clearly described?

-Do the authors discuss how these data can be helpful to advance our understanding of the topic under study?

-Is public health relevance addressed?

Reviewer #1: Yes, conclusions are based on results of the modelling protocol. The results are interesting and portray possible positive impact of climate change on tsetse-related human and animal health.

Reviewer #3: The conclusions drawn have been largely based on the data obtained and the observation that the data used is limited. It is encouraging to note that the authors concede the matter of data size or period of study as a limitation. The authors have suggested various scenarios where the out can be used in a manner that benefits different categories of people.

**Editorial and Data Presentation Modifications?**

Reviewer #1: (i) Line 66-68: 'For instance, trypanosomiasis in livestock causes loss of over 4 billion USD due to 70% reduction of cattle density, 50% reduction in diary'... the authors must have meant 'dairy', not 'diary'.

(ii) Line 85: 'Specifically, these models relies on statistical correlations between species'....'rely' instead of 'relies'?

(iii) Line 135-6: 'Since blood meals is the only known tsetse fly nutrition, no information is currently known on effects of precipitation on tsetse fly species except reports that indicate fluctuation of abundance during rainy season'......'available' instead of 'known'?

(iv) Can the authors provide the rationale on the deployment of RCP 4.5 stabilization scenario?

Reviewer #3: After going through the manuscript, my overall impression is to accept the paper as it demonstrates a method of prediction that may be useful to tsetse researchers and control. There are some minor typing errors which I believe the Authors may find when they proof read their manuscript.

**Summary and General Comments**

Reviewer #1: In addition to the few suggestions made above under 'Editorial and Data Presentation Modifications', I think the authors need to go through the manuscript carefully and make some editorial changes or improvements.

Reviewer #3: The manuscript has focussed on demonstrating the reliability or accuracy of one method of predicting the distribution of three species of tsetse flies prevalent in an area where risk of contracting disease is high for both animals and the people. The study attempts to address a real problem by providing a cost reducing means for planners. The key factors of temperature as minimum and maximum, altitude and indirectly rainfall that have been used in the study are already used in planning and the tool may help to reduce planning costs by enhancing the likely places with tsetse flies in an area. This is a modified method that has been used before and it is more refined. The study demonstrates the feasibility for use despite having been based on data from a short period of study. The methodology has no direct involvement of humans and animals but one hopes that the team responsible for field data collection was made aware of the risks associated with the study and that they were sufficiently protected as field staff.

PLOS authors have the option to publish the peer review history of their article (what does this mean?). If published, this will include your full peer review and any attached files.

Reviewer #1: No

Reviewer #3: No

---

## [Editor Report · Acceptance letter]

4 Feb 2021

Dear Dr Nnko,

We are delighted to inform you that your manuscript, "Potential Impacts of Climate Change on Geographical Distribution of Three Primary Vectors of African Trypanosomiasis in Tanzania Maasai Steppe; *G. m. morsitans, G. pallidipes and G. swynnertoni*," has been formally accepted for publication in PLOS Neglected Tropical Diseases.

Best regards,

Shaden Kamhawi

co-Editor-in-Chief

Paul Brindley

co-Editor-in-Chief
